# JUDGE DECODING: FASTER SPECULATIVE SAMPLING REQUIRES GOING BEYOND MODEL ALIGNMENT

**Gregor Bachmann**[†,§,*]  **Sotiris Anagnostidis**[†,§]  **Albert Pumarola**[†]

**Markos Georgopoulos**[†]  **Artsiom Sanakoyeu**[†]  **Yuming Du**[†]

**Edgar Schönfeld**[†]  **Ali Thabet**[†]  **Jonas Kohler**[†]

## ABSTRACT

The performance of large language models (LLMs) is closely linked to their underlying size, leading to ever-growing networks and hence slower inference. Speculative decoding has been proposed as a technique to accelerate autoregressive generation, leveraging a fast draft model to propose candidate tokens, which are then verified in parallel based on their likelihood under the target model. While this approach guarantees to reproduce the target output, it incurs a substantial penalty: many high-quality draft tokens are rejected, even when they represent objectively valid continuations. Indeed, we show that even powerful draft models such as `GPT-4o`, as well as human text cannot achieve high acceptance rates under the standard verification scheme. This severely limits the speedup potential of current speculative decoding methods, as an early rejection becomes overwhelmingly likely when solely relying on alignment of draft and target.

We thus ask the following question: *Can we adapt verification to recognize correct, but non-aligned replies?* To this end, we draw inspiration from the *LLM-as-a-judge* framework, which demonstrated that LLMs are able to rate answers in a versatile way. We carefully design a dataset to elicit the same capability in the target model by training a compact module on top of the embeddings to produce "judgements" of the current continuation. We showcase our strategy on the `Llama-3.1` family, where our `8b/405B-Judge` achieves a speedup of $9\times$ over `Llama-405B`, while maintaining its quality on a large range of benchmarks. These benefits remain present even in optimized inference frameworks, where our method reaches up to 141 tokens/s for `8B/70B-Judge` and 129 tokens/s for `8B/405B` on 2 and 8 H100s respectively.

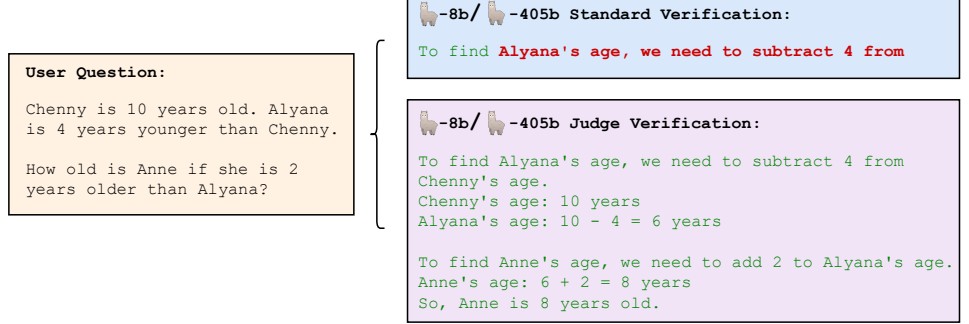

Figure 1: Standard speculative decoding versus our judge decoding strategy for `Llama-3.1-8B` as draft and `Llama-3.1-405B` as target. Accepted (rejected) tokens are highlighted in green (red).

[*]Work done during an internship at Meta GenAI. [†]Meta GenAI. [§]ETH Zürich.
Correspondence to: `gregorb@ethz.ch`

# 1   INTRODUCTION

Large language models have transformed the field of natural language processing in recent years, displaying astounding capabilities across various tasks (Radford et al., 2019; OpenAI et al., 2024). The performance of these models is closely tied to their underlying size, with bigger models often achieving significantly better results across benchmarks (Kaplan et al., 2020; Hoffmann et al., 2022). For example, Meta recently released their largest and best model to date with `Llama-3.1-405B`, boasting an enormous parameter count of 405 billion (Dubey et al., 2024).

While offering great performance, such big models require a vast amount of resources to be deployed, and inference efficiency starts to pose a critical problem. Due to the autoregressive nature of decoding coupled with the large parameter count, token generation becomes a memory-bound process, especially at small batch sizes (Shazeer, 2019; Ivanov et al., 2021; Pope et al., 2023). To speed up inference in such a setting, *speculative decoding* (SD) has been proposed (Stern et al., 2018; Xia et al., 2023; Chen et al., 2023; Leviathan et al., 2023), a technique that leverages the fact that processing several tokens in parallel comes at no additional latency cost in the memory-bound regime. More concretely, a small but fast *draft* model produces a sequence of $M$ candidate tokens, which are then verified in parallel by the model of interest, usually referred to as the *target* model. In standard SD, the target model accepts a candidate token if it assigns the same or higher probability given the context, otherwise a biased coin is flipped. If at least one candidate token is accepted, inference time is reduced as the large model needs to be called only once to produce multiple tokens. As shown in Chen et al. (2023), such a strategy provably preserves the distribution of the target model, while achieving significant speedups.

Relying on target probabilities guarantees the same output, but as a consequence, a token is solely judged based on its alignment with the target model, and not by its inherent contextual quality. As a consequence, current approaches set the number of candidate tokens to small numbers $M \in \{5, 7\}$, as an early rejection becomes overwhelmingly likely when drafting more. On the other hand, the quality of "*small*" language models (and thus the quality of candidate tokens) has been rapidly improving recently. `GPT-4o` and the recently introduced `GPT-4o-mini` show strong performance across many benchmarks, with *OpenAI* actively recommending these models over the more expensive `GPT-4`. Similarly, `Llama-3.1-8B` has achieved bigger gains in performance over its prior iterations, compared to the larger `Llama-3.1-70B` (Dubey et al., 2024), highlighting as well that small models are catching up. `Phi-3-mini` is another small model at "only" three billion parameters that despite its size manages to match the performance of `GPT-3.5` (Abdin et al., 2024).

While draft models are rapidly improving and providing increasingly high-quality answers, alignment-based verification fails to reflect this progress, still rejecting tokens that are not perfectly aligned with the target response (see Fig. 1 for an example). Motivated by this insight, we explore the following question in this paper:

*Can we adapt verification to assess token quality rather than alignment?*

We draw inspiration from the *LLM-as-a-judge* framework, where LLMs are used to judge the quality of other model responses to user questions (Zheng et al., 2023). These judgements exhibit very strong correlation with human ratings, making this a cheap and scalable approach for model quality evaluation. Interestingly, LLM-judges display the ability to rate answers in a versatile way, allowing them to appreciate correct but potentially unaligned responses. To equip the target model with similar capabilities, we design a small dataset consisting of correct and wrong replies to user questions. We create a diverse set of responses from several models and precisely annotate the location of the mistaken tokens. We then leverage the powerful target embeddings to train a small module with the objective of predicting the correctness of a given token, mimicking the LLM-judge mechanism.

In summary, we make the following contributions:

- We demonstrate through a series of experiments how the decision mechanism in speculative decoding rejects many high quality tokens, identifying a key limitation of the technique.

- We adapt verification using ideas from *LLM-as-a-judge*, eliciting the same versatile rating capability in the target by adding a simple linear layer that can be trained in under 1.5 hours.

- Using a `Llama 8B/70B-Judge`, our approach obtains speedups of $9\times$ over standard decoding, achieving an unprecedented 129 tokens/s, while maintaining the quality of `Llama-405B` on a range of benchmarks.

## 2 RELATED WORKS

Speculative decoding has been used and extended in a range of works, leading to significant speedups across many model families and datasets. Several ideas for draft models have been explored in the literature. Early papers rely on specialized draft models (Sun et al., 2021; Xia et al., 2023) or smaller versions of the target model (Chen et al., 2023; Leviathan et al., 2023), usually trained using the same data and learning protocol. Another line of work uses shallow networks on top of the target embeddings as a drafter with the goal to predict multiple tokens into the future (Stern et al., 2018; Cai et al., 2024; Li et al., 2024a;b; Zhang et al., 2024a; Wertheimer et al., 2024; Ankner et al., 2024; Gloeckle et al., 2024; Bhendawade et al., 2024). Other approaches use a sub-network of the target model, e.g. Schuster et al. (2022); Zhang et al. (2024b); Elhoushi et al. (2024); Liu et al. (2024b;a) skip a percentage of the layers to produce candidate tokens. Other architectures have also been explored: Wang et al. (2024) develop a SD variant for `Mamba` models (Gu & Dao, 2024), while Christopher et al. (2024) explore diffusion-based language models as drafters. Other components of the process have been investigated as well; Huang et al. (2024); Liu et al. (2024c) analyze the number of draft tokens $M$ with the goal of learning to choose it dynamically. Kim et al. (2023) take a similar approach aiming to measure the uncertainty of the draft model, allowing the target to take over when needed. Monea et al. (2023); Bachmann & Nagarajan (2024) on the other hand explore parallel decoding without a draft by conditioning on "look-ahead" tokens or using Jacobi iterations (Santilli et al., 2023; Fu et al., 2024), allowing the target to produce several tokens in one step.

Many works have aimed to improve the acceptance rates in SD: Zhou et al. (2024) encourage higher alignment by finetuning with a distillation loss, Li et al. (2024a); Cai et al. (2024); Ankner et al. (2024); Miao et al. (2024); Chen et al. (2024) construct token trees out of the top-$K$ predictions in various ways and verify them in parallel using tree-attention, covering thus a larger space of token combinations. Other methods propose to exchange more information between draft and target: Du et al. (2024) allow the draft to access the key-value cache of the target, while in S et al. (2024) the draft is further conditioned on target activations. All these methods improve acceptance rates by either encouraging better alignment with more information or producing more guesses in parallel. This is different from our work, which seeks to change the verification scheme itself.

## 3 VERIFICATION IN SPECULATIVE DECODING

### 3.1 BACKGROUND

**Speculative decoding.** Denote by $\text{LLM}_{\text{targ}}$ and $\text{LLM}_{\text{draft}}$ the target and draft model respectively. We use $\mathcal{V} = \{1, \dots, V\}$ for the vocabulary. Let $M \in \mathbb{N}$ represent the number of candidate tokens, $m_*$ the number of accepted tokens and $\boldsymbol{s} \in \mathcal{V}^L$ the context. Let us denote by

$$(t_1, \boldsymbol{p}_1), \dots, (t_m, \boldsymbol{p}_m) = \text{LLM}^{(m)}(\boldsymbol{s}) \tag{1}$$

an autoregressive sampling of $m$ tokens from $\text{LLM}$ given context $\boldsymbol{s}$, where $t_1, \dots, t_m \in \mathcal{V}$ are the sampled tokens and $\boldsymbol{p}_i \in \mathbb{R}^V$ the corresponding softmax probabilities. Further, we denote by

$$\boldsymbol{p}_1, \dots, \boldsymbol{p}_{m+1} = \text{LLM}(t_1, \dots, t_m; \boldsymbol{s}) \tag{2}$$

running the (parallel) forward pass of $\text{LLM}$ on tokens $t_1, \dots, t_m$. Notice that this produces one more probability vector $\boldsymbol{p}_{m+1}$ as we now also process token $t_m$.

In SD, the draft model autoregressively produces $M$ candidate tokens given the current context $\boldsymbol{s}$ using any sampling scheme (but usually greedy),

$$(c_1, \boldsymbol{q}_1), \dots, (c_M, \boldsymbol{q}_M) = \text{LLM}_{\text{draft}}^{(M)}(\boldsymbol{s}) \tag{3}$$

where $c_1, \dots, c_M$ are the sampled candidate tokens and $\boldsymbol{q}_i \in \mathbb{R}^V$ for $i = 1, \dots, M$ are the corresponding probability vectors over the vocabulary. The probability of token $c_i$ under the draft model is thus $\boldsymbol{q}_i[c_i]$. The target model then processes these tokens in parallel, resulting in probability vectors $\boldsymbol{p}_1, \dots, \boldsymbol{p}_{M+1} = \text{LLM}_{\text{targ}}(c_1, \dots, c_M; \boldsymbol{s})$. Rejection now works as follows:

$$\text{Accept } c_i \text{ if all previous tokens are accepted and } \epsilon_i < \frac{\boldsymbol{p}_i[c_i]}{\boldsymbol{q}_i[c_i]} \text{ for } \epsilon_i \sim \mathcal{U}\left([0, 1]\right) \tag{4}$$

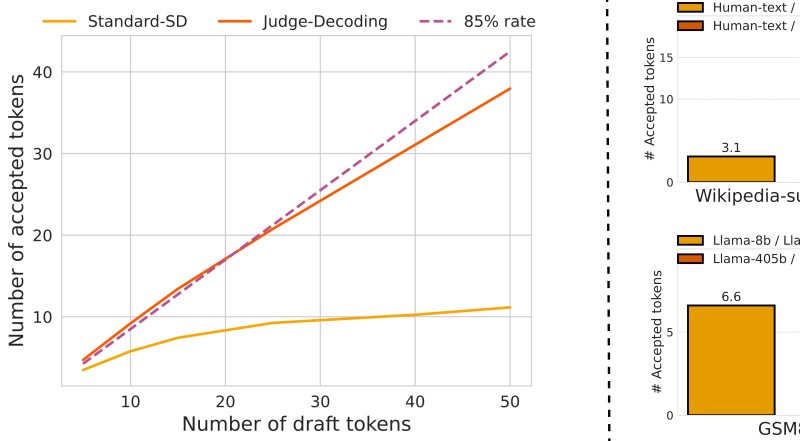

Figure 2: **Left:** Average number of generated tokens as a function of the number of draft tokens $M$ for `Llama-8B/405B` with standard and judge verification. **Right:** Number of accepted tokens on high-quality human text (top) and for both `8B/405B` and `405B/8B` (bottom), both standard SD.

In words, a candidate $c_i$ is accepted if the probability under the target model is even larger. If the probability is smaller, a stochastic decision is made according to the discrepancy between the probabilities. Crucially, one is always guaranteed to produce at least one valid token: $p_1$ is solely a function of the current context $s$ and can thus be used to sample a token according to the target distribution. Similarly, if $c_i$ is the first rejected token, one can sample a correct token from $p_i$. Finally, when all candidate tokens are accepted, an extra token can be sampled from $p_{M+1}$. The accepted tokens are then added to the current context $s$ and we repeat the steps until completion.

**Number of draft tokens.**  An immediate question comes to mind when examining speculative decoding: How many draft tokens should one choose for optimal speedup? On the one hand, if the draft model produces good tokens, one would ideally want to draft a high number of candidate tokens $M$ to avoid invoking the expensive target model too many times. On the other hand, if many candidates end up being rejected, one wants to avoid spending the unnecessary drafting time. The ideal $M$ thus heavily depends on the acceptance rate, which in turn naturally depends on the verification scheme. In Fig. 2 we plot the average number of accepted tokens as a function of $M$ for the model pair `Llama-3.1-8B` and `Llama-3.1-405B` evaluated on `MT-Bench` (Zheng et al., 2023) and `GSM8K` (Cobbe et al., 2021) (yellow curve). We observe that the number of accepted tokens quickly saturates as a function of $M$ and the acceptance rates thus decrease rapidly. As a result, choosing a large number of draft tokens $M$ solely calls the draft model more in vain, leading to inefficient inference overall. This is the reason why prior work is limited largely to $M \leq 7$.

### 3.2 Limitations of Standard Verification

**Rejected tokens.**  What types of tokens get rejected in such a setup? In order to obtain an intuition, we explore the behaviour of SD on several benchmarks including `GSM8K`, `MT-Bench` and `HumanEval` (Chen et al., 2021). We use `Llama-8B` as the draft and `Llama-405B` as the target model. While there is a significant discrepancy in terms of performance between these two models, it is worth highlighting that the draft model achieves competitive scores on all these tasks. Higher acceptance rates would thus not necessarily reduce the quality of the output on many of these examples. In fact, a large number of draft answers could be accepted as they are, especially those addressing relatively simple queries.

Notably, even in instances where the draft model produces entirely accurate solutions, the target model frequently rejects numerous tokens due to the stringent nature of the verification process. This rejection occurs despite the correctness of the solution, as the target model seeks alignment with its own response rather than contextual accuracy (Liu et al., 2023). As illustrated in Fig 1, a correct answer can be rejected after only two tokens, underscoring the potential for relaxed verification

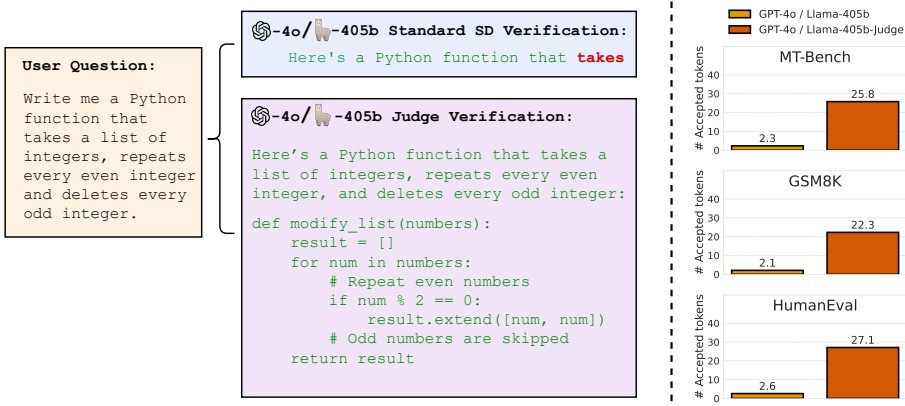

Figure 3: **Left:** Standard SD and our judge decoding when `GPT-4o` is drafting and `Llama-405B` is verifying. Green denotes accepted and red rejected tokens. **Right:** Number of accepted tokens for `GPT-4o` as draft and `Llama-405B` as target for standard speculative and our judge verification.

schemes.[1]  Intuitively, one would expect from a well-calibrated verification scheme to allow for accepting candidate tokens whenever they are contextually correct. However, as we show in the following two paragraphs, this not the case for standard logits-based verification.

**High-quality draft model.**    To further demonstrate how valid responses incur high rejection rates, we perform the following experiment: We take a very powerful LLM as the draft model and evaluate whether the target model accepts more candidate tokens, which are now guaranteed to be of high quality. While such a setup does not make sense for SD from an efficiency point of view (a powerful drafter is of course too slow), it further investigates if acceptance rates improve with the quality of responses. To that end we use `GPT-4o` as draft model for the target `Llama-405B`. We generate full answers with `GPT-4o` on `MT-Bench`, `GSM8K` and `HumanEval` and simply check how many tokens the target accepts under greedy decoding before the first rejection, as there is no way to properly perform SD with closed-source models. In order to ensure that the target model is able to "recognize" the high-quality tokens, we use the performant `Llama-405B`. We display the average acceptance length and an example prompt in Fig. 3. Counter-intuitively, we find that the target model does not reward the higher quality of tokens, accepting only roughly two before encountering the first rejection. To further explore if this observation changes when running the complete process of SD, we reverse the roles of our standard setup and use `Llama-405B` as draft for a `Llama-8B` target model. Similarly, we find that reversing the roles reduces the number of accepted tokens slightly (see Fig. 2, right side), even though they are of better quality now. We thus conclude that acceptance rates do not improve with the quality of the responses.

**Human expert drafting.**    Finally, we evaluate the efficacy of human annotations as candidate tokens for `Llama-405B` by processing *Wikipedia* articles. Using a subset of the `wikipedia-summary` dataset (Scheepers, 2017), which contains high-quality, community-reviewed abstracts, we assess token acceptance rates under greedy SD verification when prompting the model to summarize these articles. As illustrated in Fig. 2 (right), a substantial proportion of tokens face rejection, even within this high-quality context.

In summary, we conclude that SD verification in its current form is highly inefficient, as large portions of correct answers are rejected. Motivated by this insight, the following section presents a more effective verification scheme that goes beyond model alignment in order to increase efficiency.

---

[1]Additional examples of this phenomenon are provided in Appendix C.1 for further examination.

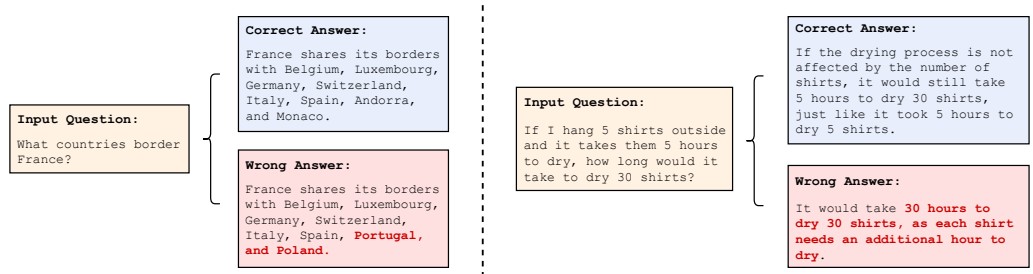

Figure 4: Two examples from our dataset. We highlight the incorrect tokens in the wrong answer in red.

# 4 JUDGE DECODING

As demonstrated by previous experiments, we need a more flexible method of verifying sequences to increase the number of accepted draft tokens, especially as draft models continue to improve in quality. Recent work by Zheng et al. (2023) showed that large LLMs can reliably act as judges to evaluate responses generated by less capable models, correlating highly with human ratings. This judging approach allows for more versatile evaluation, focusing on correctness and contextual quality rather than strict alignment. However, using LLM-judges directly is not feasible because (a) they require lengthy system prompts and often chain-of-thought reasoning, slowing inference, and (b) they evaluate full answers, whereas SD requires evaluating short, sometime partial continuations.

We thus aim to achieve this judge-like behavior efficiently while retaining the advantages of the original verification method, which ensures accurate next-token predictions in case of rejection. Since this involves computing embeddings for each draft token, we explore whether these embeddings contain sufficient information to enable rapid, reliable judgments.

## 4.1 VERSATILE AND ACCURATE VERIFICATION WITH TOKEN EMBEDDINGS

**Token embeddings signal errors.** Contrary to standard SD, which accepts or rejects a given token based on its softmax probabilities (see Eq. 4), we find that the model's reaction to processing the incorrect token itself reveals surprisingly valuable information. Specifically, our experiments show that last hidden layer embeddings of erroneous tokens effectively "flag" errors and contradictions, prompting the model to generate subsequent tokens that attempt to correct the mistake. This phenomenon is strikingly illustrated in Fig. 5, where we condition `Llama-405B` on wrong replies (highlighted in red) and observe the model's immediate efforts to rectify its response (highlighted in green). For instance, when forced to start with the incorrect statement "The capital of France is Berlin", the model continues with "... just kidding, it's actually Paris". More such examples can be found in Appendix C.3. This unexpected behavior suggests the feasibility of leveraging the embedding of the *current* token as a means of verifying its correctness. In fact, we will show in the following that a simple logistic regression head on top of these embeddings achieves high accuracy and can be trained in under 1.5 hours. Prior, embeddings have also been used to discover latent knowledge in LLMs and edit them (Burns et al., 2023; Zou et al., 2023a; von Rütte et al., 2024), further underscoring their richness.

**Dataset curation.** In order to leverage token embeddings for verification, we carefully craft a dataset consisting of high-quality user inputs, along with a correct and wrong answer pair. The set of input prompts are a mixture of newly-created questions and two public datasets that we heavily filtered (`Alpaca` (Taori et al., 2023) and `ARC` (Clark et al., 2018)). Importantly, we only use the input questions and none of the answers. We leverage several models to produce a diverse set of correct and wrong answers, including `Mistral-Large-2`, `Llama-8B` and `Llama-405B`, thereby fostering robustness of the trained judge to recognize correct but differently aligned solutions. All answers were manually reviewed and corrected by the authors, who also annotated the precise lo-

cation of errors in wrong answers[2]. In total we collected $500$ high-quality question, correct answer, wrong answer tuples. For training, we label every token from the correct answer as positive, every token from the wrong answer up until the point of mistake as positive, and finally every mistaken token as negative. Two examples from our dataset are depicted in Fig. 4. Naturally, the dataset exhibits a strong imbalance, leading to roughly $20\times$ more positive than negative examples.

**Model design and training.** Equipped with the dataset, we train a linear head $f_{\text{judge}}$ on top of the target embeddings, using a weighted cross entropy loss as the objective to counter the imbalance in the dataset. We place larger weight on the negative examples in order to ensure that the resulting judge does not falsely accept wrong tokens to limit quality degradation and perform early-stopping to reduce overfitting. We tune all hyperparameters on a small test split. We experiment with embeddings from several layers and find that deeper layers perform best with only insignificant differences, while too shallow layers are clearly worse, consistent with similar observations in previous works (Zou et al., 2023b; Gurnee & Tegmark, 2023; von Rütte et al., 2024). For simplicity, we thus stick to using the last embedding of the target before the RMS normalization (Zhang & Sennrich, 2019) and the language modelling (LM) head. While we experimented with more complex architectures, including MLPs and shallow Transformer networks, a simple linear head proved most effective, demonstrating excellent performance without overfitting. This linear classifier offers significant practical advantages: we train only 16.4k parameters on just 30k tokens in less than 1.5 hours, with all target model parameters remaining frozen. Additional details are provided in Appendix B.1.

**Inference.** How is the judge head now combined with the standard elements of SD? In essence, we use $f_{\text{judge}}$ as an additional evaluator for a given token $c_i$ (or rather its embedding $e_i \in \mathbb{R}^D$) and accept it if $\sigma\left(f_{\text{judge}}(e_i)\right) > \delta$ for $\delta \in [0.5, 1]$, where $\sigma$ is the sigmoid function. In other words, $\delta$ serves as a threshold for the confidence of acceptance and practitioners can thus choose how much to trust the judge layer. In practice we observed that there is no need to tune this value to ensure quality and leave it at the natural value $\delta = 0.5$. Given a sequence of candidate tokens $c_1, \dots, c_M$, we thus get two accept/reject masks from the target model: $z_{\text{stand}} \in \{0, 1\}^M$ as in standard SD verification and $z_{\text{judge}} \in \{0, 1\}^M$ from the judge head. We take the logical OR between the two, $z = z_{\text{stand}} \vee z_{\text{judge}}$, since when the judge rejects and standard SD accepts, the corrected token according to the target will exactly be the same token. We can thus already accept it. We illustrate this mechanism in more detail in Appendix B.2. Note that $\delta = 1$ reduces to SD.

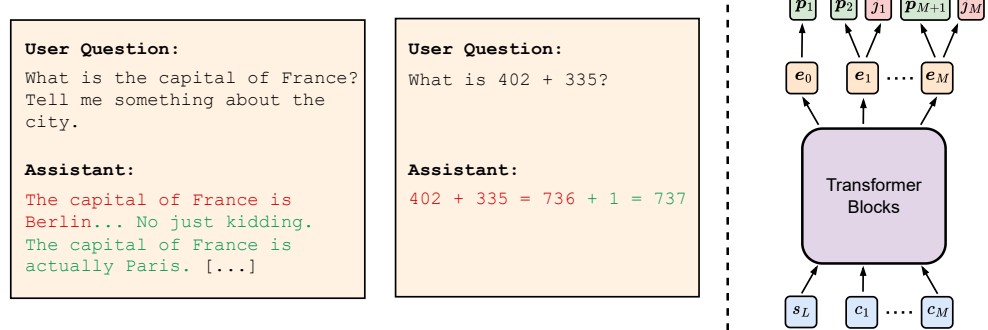

Figure 5: **Left:** Conditioning `Llama-405B` on wrong outputs. The part of the assistant response in red was forced, while parts in green were generated freely. **Right:** Judge illustration where $s_L$ is the last token from the context $s$ and $c_1, \dots, c_M$ are candidate tokens. Orange denotes embeddings, green denotes the LM-head output and red denotes the produced judgements.

---

[2]Using LLMs to that end proved to be too imprecise, which is consistent with recent observations in Tyen et al. (2024).

Table 1: Average acceptance length ($m_*$) and speedup factor over standard decoding in HuggingFace and *gpt-fast* for batch size 1. We report generation tokens/s for *gpt-fast* for 512 input and output tokens, quantized to 8-bit. All 70B (405B) models run on 2 (8) H100 GPUs, except for *Medusa (Nvidia, 2024), which runs on significantly faster H200s with NVLink Switch and TensorRT.

|  | $m_*$ | HuggingFace | GPT-Fast | Tokens/s $(512 + 512)$ |
|---|---|---|---|---|
| 8B/70B-Standard | 6.4 | 1.5× | 1.7× | 76.7 |
| 8B/70B-Judge (ours) | 18.8 | 2× | **3×** | **141.8** |
| 70B-Eagle-2 | 4.5 | **3.3×** | 1.9× | 88.1 |
| 8B/405B-Standard | 6.3 | 5.3× | 1.78× | 58.7 |
| 8B/405B-Judge (ours) | 19.7 | **9.7×** | **3.9×** | **129.3** |
| 405B-Medusa | $< 6$ | $< 6×$ | 1.9× | 108* |

## 5 Evaluation of Judge Decoding

First, we revisit the initial experiments outlined in Sec. 3.2 where we use GPT-4o as the draft model, as well as human generated text as candidate tokens. In both cases, the average number of accepted tokens is significantly higher for our method across datasets (see Fig 2 and 3). The example prompt in Fig. 3 (left) shows that the correct response of GPT-4o is fully accepted by judge decoding while standard SD rejects after two words. This illustrates that our verification scheme offers more versatile decisions.

### 5.1 Performance Benchmark

We now evaluate our verification method on standard benchmarks in the SD literature, including GSM8K (Cobbe et al., 2021), HumanEval (Chen et al., 2021) and MT-Bench (Zheng et al., 2023). In contrast to standard SD works, we do need to report the achieved accuracy values of our strategy, as adapting verification comes with the possibility of accepting wrong tokens and thus worse performance. To give a more complete picture, we further include multiple-choice benchmarks ARC (Clark et al., 2018) and MMLU (Hendrycks et al., 2021), which are atypical tasks for standard SD as only a few tokens need to be produced, but further serves as a check that our verification scheme does not degrade performance. We use the prompting templates from Dubey et al. (2024).

**Training-free baseline.** To provide more context for our results and to demonstrate that our judging strategy goes beyond simple heuristics, we also explore a simple training-free method to relax the acceptance scheme. In particular, we investigate top-K verification, where a candidate token $c_i$ is accepted, if it is among the $K$ highest valued probabilities $\boldsymbol{p}_i$ produced by token $c_{i-1}$. $K \in \mathbb{N}$ is a hyper-parameter of the decoding technique that trades-off quality against speed. Setting $K = V$ reduces to running just the fast draft model, while $K = 1$ results in standard SD.

**Preserving target performance.** We display the accuracy of *judge decoding* alongside the vanilla draft and target models, as well as top-$K$ decoding in Fig. 6 for both Llama-405B and Llama-70B. We observe that *judge decoding* almost exactly preserves target performance for all benchmarks, showing hence that up to $\sim 20$ tokens can be accepted on average from modern draft models without loss of quality. The simple heuristic baseline, on the other hand, is hardly able to improve over the draft model (even for $K = 5$), highlighting the difficulty of the problem we address with the learned head.

### 5.2 Speed Benchmark

The end-to-end speed-ups achieved by SD methods improve as mainly two factors increase: (1) the number of accepted tokens and (2) the latency gap between draft and target model. Importantly, the latter is heavily dependent on whether or not orthogonal inference time optimizations like quantization, model parallelism and graph/kernel optimization techniques (like torch.compile and TensorRT)

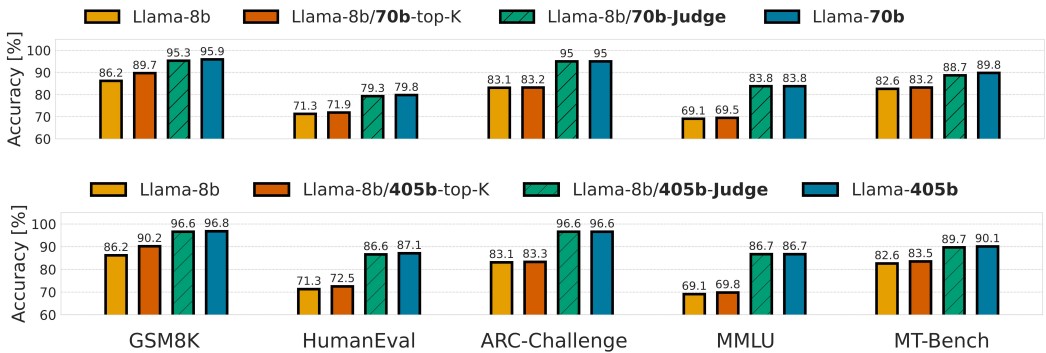

Figure 6: Benchmark results. **Top:** Draft `Llama-8B` and target `Llama-70B`. **Bottom:** Draft `Llama-8B` and target `Llama-405B`. We show top-$K$ decoding, standard SD for $M = 10$ and our judge decoding for $M = 25$ (striped). Notice that our judging method preserves accuracies very well, while top-$K$ loses most performance.

are applied. Unfortunately, prior works on SD have almost excursively relied on the user-friendly – but un-optimized – library HuggingFace (Wolf et al., 2020) to implement their methods. Yet, as rightfully pointed out by (Wertheimer et al., 2024), speed-ups of prior SD methods reported in HuggingFace tend to shrink significantly when moving to optimized inference frameworks. For example, the acceleration of `Eagle` on `Llama-2-7B` reduces from $3\times$ to merely $1.5\times$ when using *gpt-fast* as reported in Li et al. (2024a). In fact, vanilla `Llama-70B` *without any speculative decoding* achieves a higher throughput in *gpt-fast* than the state-of-the-art SD method `Eagle-2` does on the same model in HuggingFace ($\sim45$ vs $\sim33$ tokens/s).

To offer a complete picture, we here provide latency benchmarks in both frameworks, HuggingFace to facilitate comparison, as well as the arguably more relevant and optimized *gpt-fast* framework (Pytorch-Team, 2023). If not stated otherwise, we run `Llama-70B` and `Llama-405B` on 2 and 8 Nvidia H100 GPUs respectively. Our results are summarized in Table 1.

**Llama-3.1-70B.** When drafting with `Llama-8B` for `Llama-70B` with batch size 1 in simple frameworks, the latency delta between the two models is relatively small, limiting the speed-ups of *judge decoding*. This is particularly evident when compared to SD methods that leverage small LM heads as draft modules (such as `Eagle-2` Li et al. (2024b) and `Medusa` Cai et al. (2024)). However, in the more realistic setting of deployment within an optimized inference framework, several latency bottlenecks (like CPU instruction and memory I/O) are alleviated, resulting in a more pronounced latency delta between the target and draft models. Consequently, our method effectively capitalizes on this increased latency disparity and outperforms the current state-of-the-art by a substantial margin (see right-hand side of Table 1).

**Llama-3.1-405B.** Replacing the target model with the more powerful `Llama-405B` model significantly increases verification latency. As a result, drafting (and accepting) longer sequences becomes more crucial for the overall runtime. In such settings, *judge decoding* shines because the average number of accepted tokens is $> 3\times$ larger than prior works (left-hand side of Table 1). In particular, both `Medusa` and `Eagle-2`[3] are limited to drafting $\leq 6$ token at the time, by the number of heads and the draft tree depth respectively. Our `8B/405B-Judge`, however, accepts close to 20 tokens at a time and thereby achieves a $9.7\times$ speed-up in HuggingFace and unprecedented 129 tokens/s in *gpt-fast*.

### 5.3 OUT-OF-DISTRIBUTION PERFORMANCE

Finally, we investigate to what degree our judge-decoding strategy extends to situations for which it has not been trained. To this end, we filter our dataset by removing all coding examples, train the verification head for `Llama-405B` on this reduced set and then evaluate on the coding task

---

[3]of which no 405B version exists.

`HumanEval`. While we do observe a drop in performance from $86.6$ to $80.4\%$, the performance is still significantly better than the draft model at $71.3\%$, indicating that the notion of "correctness" transfers between tasks at least to some degree. Nevertheless, our approach is not a silver bullet and to maintain target quality it is required to train the judge on data of similar nature.

## 6 Conclusion

In this work we have investigated the verification mechanism in speculative decoding and identified how its focus on alignment between draft and target response leads to the rejection of objectively correct continuations. To fully leverage the improved quality of "small" language models, we thus proposed an adapted verification scheme that makes use of the capability of LLMs to judge responses in a versatile way. This allows for efficiently drafting more tokens, leading to significant speedups up to $9\times$ on a range of benchmarks, achieving unprecedented speeds of 129 tokens/s for `Llama-405B`. In the regime of many draft tokens, "small" language models shine as drafters compared to the small modules employed in approaches like `Eagle` or `Medusa` and we believe this trend will only further accentuate in the future. Our approach however also comes with a drawback; the mathematical guarantee to maintain target quality is lost by relying on the judge. Through extensive experiments we show that a well-trained judge does not lose performance on standard benchmarks and we thus view our approach as a significant first step into this direction. On the other hand, our strategy in its current version does not present a silver bullet; novel tasks require the careful annotation of similar data to maintain quality, otherwise performance is lost. The small amount of data required in our setup is nevertheless a very encouraging sign. Future work can hopefully build upon our contributions, further improving our judge decoding strategy to enable more speedups.

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

## A    LIMITATIONS

Here we list limitations of our approach to the best of our knowledge:

- An obvious limitation is the loss of the mathematical guarantee to match target quality. While we perform extensive experiments and show that quality is maintained, there is no certainty for novel tasks.

- The draft model needs to be of high-quality, otherwise our approach naturally does not prove beneficial and too many tokens end up being rejected. Self-speculation and small drafters in the spirit of `Medusa` or `Eagle` are thus not ideal since their generations quickly deteriorate when drafting too far into the future.

- Similarly, the target model needs to be of sufficient size to be able to provide accurate judgements. Speedups for smaller models such as `Llama-8B` are hence tougher to achieve.

- As highlighted in the main text, new tasks do require careful annotation of data to maintain quality. The required amount on the other hand turns out to be small in our case.

- If the draft model has safety issues, the target model could potentially accept safety-critical tokens through the judge, even if the target would otherwise never produce such outputs. We have not observed such issues in our experiments but have also not thoroughly investigated this problem as it is beyond the scope of our work.

## B    ARCHITECTURAL AND EXPERIMENTAL DETAILS

### B.1    LINEAR HEAD

We train our linear heads using the `AdamW` optimizer (Loshchilov & Hutter, 2019) with learning rate $\eta = 0.0001$, weight decay $0.1$ and batch size $128$. Note that for `Llama-405B`, our linear head has dimension $16,384$ while for `Llama-70B` it has $8,192$. Our linear head can be viewed as an additional entry in the vocabulary $\mathcal{V}$, reducing its inference overhead thus to practically zero.

### B.2    JUDGE MASKING

We describe the the combination of standard and judge mask in more detail in Fig. 7. In the following

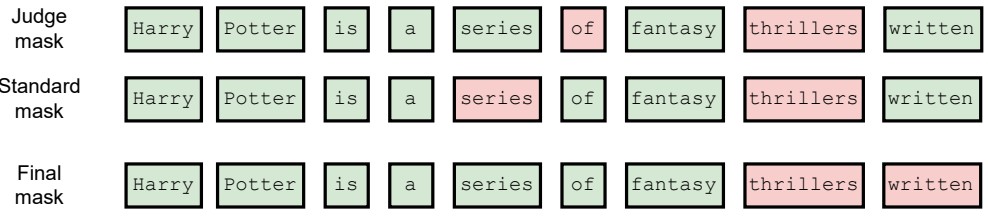

Figure 7: Illustration of mask creation in judge decoding. The decision mask resulting from the judge is combined with the standard mask from SD. Once both methods disagree, subsequent tokens get rejected automatically as usual, even if they were individually accepted.

we will describe in more detail why it is natural to combine the masks of judge decoding and standard speculative decoding. We illustrate this in Fig. 8. There are (rare) scenarios where a candidate token is rejected by judge decoding (such as "that" in the example) and the "corrected" token according to the target model happens to be the same token ("that" in blue). This situation could repeat; the very next token could again be rejected by JD and accepted by the target (token "guy" in the example). Standard SD on the other hand would accept all those tokens as the draft exactly matches the target suggestion. We eventually end up accepting the exact same tokens (in case of rejection we have to trust the target), so it makes sense to combine the masks and use the SD mask to not end up repeating the steps (in the example we combine steps 1., 2., 3. into one step on the right). In our experiments we do not observe this situation too often, but it can occasionally occur as the judge was tuned to rather reject than accept when in doubt to avoid false positives.

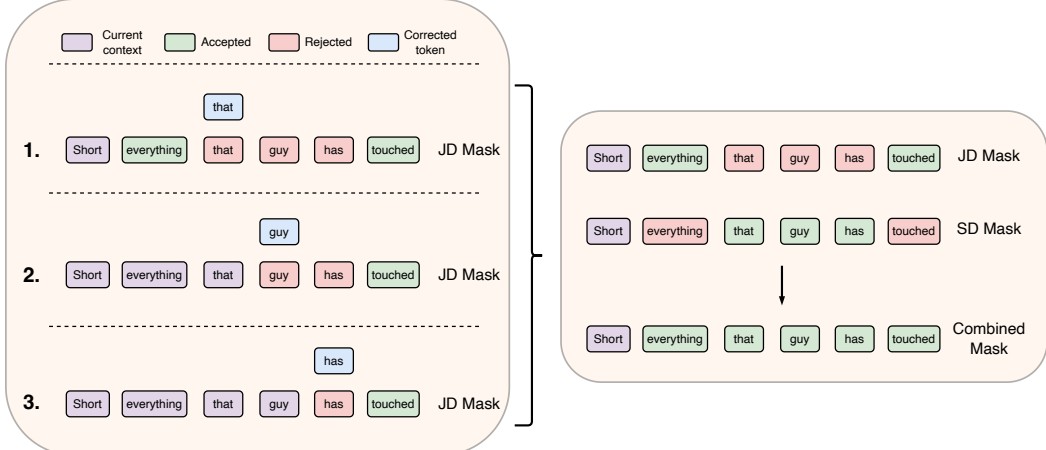

Figure 8: Illustration how combining the masks of judge decoding (JD) and standard speculative decoding (SD) results in the same reply but in less steps in the (rare) case that JD rejects a token that is actually the target token.

### B.3 HARDWARE

We run all of our experiments on a single node of H100-SXM5 GPUs. For `Llama-405B` we use 8 GPUs and 8-bit quantization to ensure that the model fits on a single node. For `Llama-70B`, we use again 8-bit quantization but only 2 GPUs.

## C MORE PROMPTS

### C.1 REJECTED REPLIES FOR LLAMA-8B

Here we provide more example prompts where `Llama-8B` provides completely correct answers but gets rejected early on by the target `Llama-405B`. We display the decision of our judge decoding strategy right below. To also highlight that judge decoding can catch errors and does not just blindly accept responses, we also show prompts where `Llama-8B` provides a wrong response.

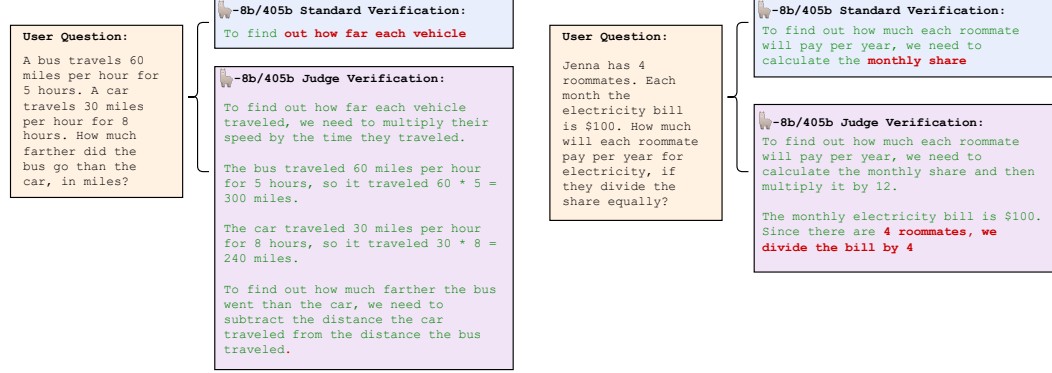

Figure 9: More example prompts for SD for `Llama-8B` and `Llama-405B`. **Left:** Correct response getting rejected early under standard decoding, while judge decoding accepts a long continuation (but admittedly over-cautiously rejects later on). **Right:** Wrong response that gets rejected too early by standard decoding and correctly rejected later on by judge decoding (there are 5 roommates in total).

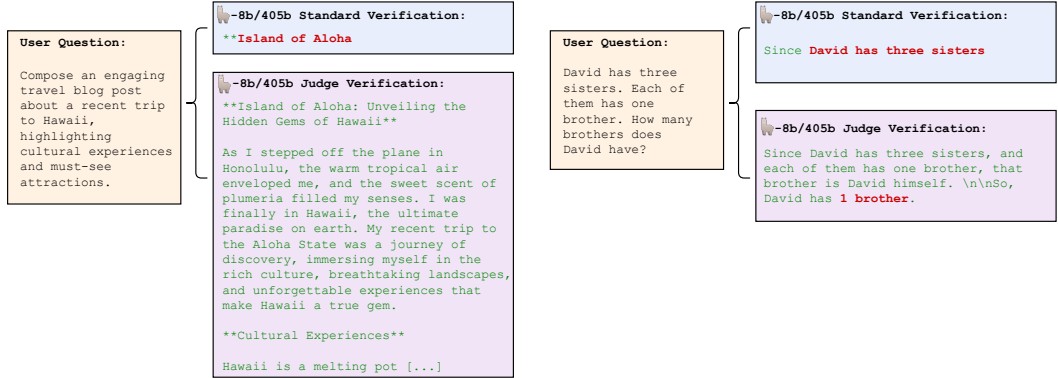

Figure 10: More example prompts for speculative decoding for `Llama-8B` and `Llama-405B`. **Left:** Correct response getting rejected early under standard decoding, while judge decoding accepts a long continuation (but admittedly over-cautiously rejects later on). **Right:** Wrong response that gets rejected too early by standard decoding and correctly rejected later on by judge decoding (there are 5 roommates in total).

## C.2 JUDGING OF WIKIPEDA ARTICLES

Here we provide more details and examples for verifying *Wikipedia* articles. Given a Wikipedia article name such as "aluminium", "Moore's Law" or "Pet Shop Boys", we prompt the model for information by asking "`What can you tell me about <insert topic>?`". We then again compare greedy matching for standard speculative decoding with our judging strategy when using the summary part of the Wikipedia article as a reply. We display some example prompts along with the corresponding verifications in Fig. 11.

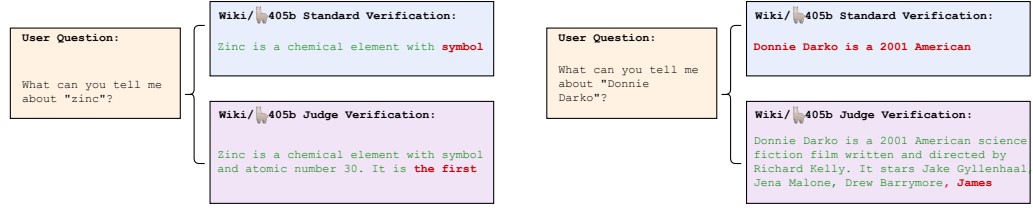

Figure 11: Two example prompts from the subset of `wikipedia-summaries`, along with the correspond verifications.

## C.3 FORCING WRONG REPLIES FOR LLAMA-405B

Here we provide some more evidence of the "correcting" behaviour of `Llama-405B` when conditioned on wrong tokens. If the model cannot fix the response anymore, then it will often point out that the completion it just gave is actually wrong, see e.g. examples in Fig. 13. This again strongly suggests that correctness should thus be detectable in the embeddings of such tokens.

```
User Question:

Write me a Python function that doubles
every element in a given list.

Assistant:

def double_elements(l):
    """Doubles every element in a list"""
    for i in range(len(l)):
        l[i] = 3 * l[i] - l[i]
    return l
```

```
User Question:

What is heavier: A kilogram of
feathers or a kilogram of stones?

Assistant:

A kilogram of stones is heavier
because... just kidding. They are
actually the same weight.
```

Figure 12: **Left:** `Llama-405B` corrects the mistake by subtracting `l[i]` to double instead of triple. **Right:** Similar correction behaviour by pointing out that response so far is wrong.

```
User Question:

Write me a Python function that doubles
every element in a given list.

Assistant:

def double_elements(l):
    """Doubles every element in a list"""
    for i in range(len(l)):
        l[i] = 2 * l[j]
    return l

However, the above function has a bug.
Here is the corrected version [...]
```

```
User Question:

Write me a Python function that sets every odd
entry in a list to zero and appends the result to
the original unmodified list.

Assistant:

def odd_to_zero(l):
    """Set every odd entry in a list to zero and
append it to the original unmodified list"""
    copy = l
    for i in range(len(copy)):
        if copy[i] % 2 == 1:
            copy[i] = 0
    return l + copy

However, the above function will modify the
original list when it is referenced by another
variable. Here is the corrected version [...]
```

Figure 13: **Left:** `Llama-405B` correctly points out that there is a bug as the index variable "`j`"" is not defined. **Right:** The model can also catch more subtle mistakes. Here the original list also gets modified as no copy was made, leading to wrong outputs.

