# OpenReview forum: "Judge Decoding: Faster Speculative Sampling Requires Going Beyond Model Alignment"
_ICLR.cc/2025/Conference — ICLR 2025 Oral_

### Official Review · Reviewer_Sf3Q · 2024-10-30

**Soundness:** 3
**Presentation:** 3
**Contribution:** 2
**Rating:** 6
**Confidence:** 3

**Summary:**

In standard Speculative Decoding (SD), draft tokens are typically accepted or rejected during the verification phase based on whether they align with the target model distribution. However, this paper raises the concern that this verification process may be too stringent, resulting in high-quality draft tokens being rejected because they don’t align to the target distribution even though they may be valid and correct continuations. The authors try to address this concern by building upon the idea of LLM-as-a-judge and training a small judgment model that works on top of the target model latent embeddings and can be used to judge whether a draft is a correct continuation or not. Using this idea, the authors present speed ups using draft and target models in the Llama-3.1 family on a range of benchmarks and present a novel dataset (TokenCourt) used to train the target judge model.

**Strengths:**

- The authors clearly present and motivate their ideas around standard SD verification potentially being overly stringent and distinguishing between semantic correctness and target distribution “alignedness”. The motivation is straightforward to understand with the examples provided and it’s clear how standard verification can be arbitrarily strict depending on the target model resulting in less than desired inference efficiency using standard SD at times.
- The reported speedups over standard SD are significant, largely in part due to the increase in the number of accepted tokens using this new judge decoding verification process compared to standard verification. However, the authors correctly point out also that this is not necessarily an apples to apples comparison given the loss of output distribution guarantees compared to standard SD.
- The investigations into token embeddings signal errors was particularly interesting in that “incorrect” statements in text could be pinpointed based on the target model’s latent embedding because of the models tendency to try to immediately rectify its response following an inaccuracy and the resulting dataset curated based on this could prove useful to others if released

**Weaknesses:**

- Given the fact that verification is being changed so there is no longer a guarantee of output distribution match, the speedup comparisons are no longer apples to apples and have to also be paired with quality comparisons to ensure quality is maintained. Additionally, given the way the judge model is trained on “correct” and “incorrect” inputs, it seems that for novel tasks, a new tuning dataset must be hand curated for each task to train the judgment model to recognize “correctness” making the overhead of applying a technique like Judge Decoding significant compared to other SD modifications.
- The notion of “correctness” is easiest to understand in the context of text generation tasks that are Question&Answer based. However, it’s unclear how this idea would work for more open-ended creative generation tasks where “correctness” may not be as easily defined and matching the target model output distribution may be ideal.
- An additional condition to applying Judge Decoding is for the draft model to be of high-quality that can produce longer valid generations otherwise the approach may not provide that much of a speedup.
- The reported experimental results are somewhat limited such as the out of distribution performance results being restricted to only a few lines and the full results not being reported in the appendix at least in comparison to the baselines etc. Additionally, it’s unclear what task is being reported on for Table 1 whereas Figure 6 shows accuracy across a variety of tasks. It would be helpful to also see the relative speedups across each of these variety of tasks as well.
- For the most part, the authors are forthcoming about weaknesses and limitations to their judge decoding approach and many of these points will align. However, the amount of pre-requisites needed to apply the technique as well as required overhead for verification of maintained quality due to the loss of output distribution alignment are substantial.

**Questions:**

- Do the authors plan on releasing their TokenCourt dataset?
- What happens when the target LLM isn’t one of the models used to generate the correct/incorrect responses to questions in TokenCourt used to train the linear judge module? Is this a necessary requirement for this technique to work? I ask because of a scenario where one may want to re-use a curated TokenCourt dataset for a new target model from an entirely different family of models to avoid having to hand annotate again.
- Looking at the showcased TokenCourt examples, one question is if Judge Decoding may result in potentially verbose outputs. And in those cases, even though more draft tokens are being accepted and decoded per second and the full output is semantically correct, it is not as concise or clear as the original target model output if standard verification were being applied?

---

> ### Author Response · Authors · 2024-11-21
> **Author response (1)**
>
> We thank the reviewer for the very thoughtful and extensive review. We will address the weaknesses and questions in the following:
>
> 1. **Verification is changed:** We completely agree with the reviewer, and we hope that this is evident enough from our main text; speedup comparisons are not enough anymore as we (deliberately) give up on the guarantee, and quality checks need to be performed. We perform various checks by evaluating our resulting model on several standard tasks for LLMs, demonstrating that our model does maintain quality. We further agree with the reviewer that some alignment between the judge dataset and the tasks of interest is needed; if we remove coding examples, we observe a significant drop in performance as shown in the main text. On the other hand, we also observe that tasks such as “MMLU” and “MT-Bench” did not need further hand-curated prompts to maintain quality, as they seem to share enough overlap with the dataset already. We thus suspect that a large enough dataset should cover a large fraction of the prompt space, but we definitely agree that very “novel” tasks most likely require annotating new samples. We also want to point out however that the number of such examples is relatively small in our experience, ranging only from 10-20 examples.
> 2. **Open-ended tasks:** We thank the reviewer for this great insight, we did not highlight this limitation enough in the main text. Our approach relies on the fact that there are correct and wrong tokens, and it is not immediately obvious how to extend this to more ambiguous tasks such as writing, summarization, and quality of response in general. One could think about ranking responses, similar to the mechanism used in RLHF to distinguish such finer differences in quality, which would make for exciting future work. We will definitely add these points to the paper!
> On the other hand, we do want to highlight that MT-Bench does measure the quality of responses through a GPT-4 judge, which does include some more ambiguous tasks such as writing and summarizing, and we do not observe a strong regression there. We view our work as a first step towards more flexible verification, and we will definitely highlight this limitation in the main text.
> 3. **High-quality draft:** We agree with the reviewer, that a high-quality draft model such as Llama-8b that is capable of drafting longer, sensible answers is needed for our approach. Otherwise, accepting more tokens will prove detrimental. We view this limitation as less worrying for the following reasons: (1) Small models have been improving rapidly over the last year, and we expect this trend to only get stronger. This means that fast draft models of high quality will become more and more available. (2) On the other hand, flagship models will most likely continue to grow in size, making thus “better (larger) models small enough in comparison”. E.g. one could imagine Llama-70b becoming a fast enough draft model if the target were as big as Llama-”3005b”. We are hence confident that this limitation is not and will not become a crucial one in the future.
> 4. **Out-of-distribution:**  We kindly ask the reviewer to clarify what is meant by “full results not being reported”. We have added a table in Appendix B.5 that evaluates the same judge without coding examples on the other tasks and as expected, we observe roughly the same scores (except for MT-bench, which also includes a small set of coding tasks). If that was not what the reviewer meant, let us know!
> We are reporting the average speedup over GSM8k, MT-Bench and HumanEval in Table 1. We have added a table in Appendix B.4 further detailing the individual speedups. As has been observed in prior work, HumanEval (a coding task) enjoys the most speedups, followed by GSM8k and then MT-Bench.
> 5. **Amount of prerequisites:** Again, we don’t disagree with the reviewer, very new tasks will require the annotation of some examples (as we show in the paper), but we also found that number to be very small (10-20 prompts). Other tasks such as MMLU on the other hand were even already covered well by simply having other multiple choice prompts in the training set, so the concept of “correct” and “wrong” does generalize between different tasks. We thus believe that the amount of additional data needed for new tasks will shrink significantly as the training data for the judge grows.
> We further believe that the very significant speedups of our technique do justify some of the overhead, especially when the domain of application is well-known in advance.
> Finally, we would like to stress again that we don’t want to claim that our approach represents a silver bullet. To the best of our knowledge, it is the first work in this novel direction, and we are convinced that future work can build upon it and further refine it. We are happy to further stress this in the paper, but we believe that we were already very honest about the limitations, as the reviewer also points out.

---

> > ### Author Response · Authors · 2024-11-21
> > **Author response (2)**
> >
> > 6. **Release of TokenCourt:** We are still in the process of evaluating the feasibility of opensourcing the dataset. While we hope to make it available, we cannot make any guarantees at this stage.
> >
> >
> > 7. **Other target models:** We expect the dataset to generalize for target models of similar quality. The only issue we envision is when the new target model cannot answer some prompts correctly that were included in the dataset. The embeddings of those correct tokens could be less useful in that case. As soon as there is enough overlap between targets we believe that datasets should generalize. We also want to highlight that we did use two target models for the same dataset, Llama-405b, and Llama-70b, without observing major regressions, which further makes us optimistic that datasets will generalize between target models.
> >
> >
> > 8. **More verbose outputs:**  This is a very interesting and valid point raised by the reviewer. We have compared the average response length in terms of number of tokens for Llama-8b, Llama-405b and Llama-8b/405b-Judge to see whether we observe strong discrepancies in terms of length. We display a table of the results at the bottom of the response. We observe that there are no strong discrepancies between draft and target model to begin with, and the judge seems to reduce the small level of verbosity of the draft, if anything. Interestingly for HumanEval, we observe that the target produces longer outputs. This is mostly the case because it seems to produce more detailed comments and longer overall explanations of the code.
> > We agree however that if there are situations where the draft is significantly more verbose than the target, then our judge model will likely also produce more verbose outputs, but still on a smaller scale than the draft.
> > We hope that this addresses the point of the reviewer.
> >
> >
> >
> >
> > | 	 | GSM8K | HumanEval | MT-Bench
> > | --- | :-:    | :-:            | :-: |
> > | 8b       | 182       | 371	   | 425
> > | 405b   | 159       | 409	   | 387
> > | 8b/405b-Judge |  167      | 388	   | 405

---

> > > ### Author Response · Authors · 2024-11-29
> > > **Followup**
> > >
> > > We appreciate that the reviewer most likely has a busy time schedule. However, it would be nice to further discuss our work and our rebuttal with the reviewer to further improve the paper. We would appreciate any further feedback on points that remain unaddressed, if there are any left.

---

> > > > ### Comment · Reviewer_Sf3Q · 2024-11-29
> > > >
> > > > Thanks to the authors for the detailed response and commitment to improvements to the main paper. It is clear that the authors understand and are forthcoming about the limitations of their novel approach but remain enthusiastic about the potential benefits. I've updated my score accordingly.

---

### Official Review · Reviewer_h6mH · 2024-11-01

**Soundness:** 3
**Presentation:** 4
**Contribution:** 4
**Rating:** 10
**Confidence:** 4

**Summary:**

The authors propose "Judge Decoding," a novel approach to speculative sampling that challenges the fundamental assumption that verification must be based on model alignment. Instead of using likelihood comparisons, they train a simple linear classifier on the last embedding layer of the target model to judge token quality directly. Their key insight is that current speculative decoding methods reject many high-quality tokens simply because they don't perfectly align with the target model's preferred phrasing. By analyzing embeddings and observing how models attempt to correct errors, they show that correctness information is already present in the token representations. They leverage this insight to create a lightweight judge module (16.4k parameters) trained on their carefully curated TokenCourt dataset. The method achieves up to 9× speedups over standard decoding, reaching unprecedented speeds of 129 tokens/s for Llama-405B, while maintaining quality across a comprehensive range of benchmarks.

**Strengths:**

- Clever idea that was well-executed. The paper is well-written and the experiments make sense. The results are impactful in practical terms (people should use this) and for spawning future research (there are many possible variations on how to train the linear classifier).

- I liked that the authors decided to go with the simple linear classifier since it worked best, rather than overcomplicating things

- I also liked the systematic investigation with illustrative examples for how judge decoding differs from speculative decoding

- Achieves state-of-the-art performance (141 tokens/s for 8B/70B-Judge, 129 tokens/s for 8B/405B) on standard hardware (2-8 H100 GPUs).

**Weaknesses:**

I would have liked if the authors addressed the possible impact of their method on less binary (correct vs incorrect) metrics like reward model scores or chatbot ELO scores. I could imagine to get a more noticeable regression on those, since the writing style of models is more important in that setting. Perhaps something for the discussion?

The Burns et al 2022 paper (https://arxiv.org/abs/2212.03827) does some nice work on eliciting concepts with linear probes which could be relevant here.

**Questions:**

No questions

---

> ### Author Response · Authors · 2024-11-21
> **Author response**
>
> We thank the reviewer for the very positive feedback and the helpful review. We will address the questions and weaknesses in the following:
>
> 1. **Less binary tasks:** This is a great insight and we agree that we did not highlight this limitation enough in the main text. Indeed, our approach currently relies on the fact that there are correct and wrong tokens, and it is not immediately obvious how to extend this to more ambiguous tasks such as writing, summarization, and quality of response in general. One could think about ranking responses, similar to the mechanism used in RLHF to distinguish such finer differences in quality, which would make for exciting future work.
> On the other hand, we do want to highlight that MT-Bench does measure the quality of responses through a GPT-4 judge, which does include some more ambiguous tasks such as writing and summarizing, and we do not observe a strong regression there.
> In general, however, we definitely agree with the reviewer that stronger regressions are likely to be observed in such settings. We view our work as a first step towards more flexible verification, and we will highlight this limitation in the main text.
> 2. **Related works.** We thank the reviewer for sharing this relevant work, we have included it in Section 4.1.

---

> > ### Comment · Reviewer_h6mH · 2024-11-26
> >
> > Thank you for your response, I remain excited about this paper.

---

### Official Review · Reviewer_kfoq · 2024-11-04

**Soundness:** 3
**Presentation:** 4
**Contribution:** 3
**Rating:** 8
**Confidence:** 4

**Summary:**

The paper introduces Judge Decoding, a speculative decoding enhancement that improves token acceptance by evaluating contextual correctness rather than strict alignment with the target model. Traditional speculative decoding often rejects accurate tokens from draft models due to misalignments rather than incorrectness, limiting speedups. By training a lightweight “judge” module on a custom TokenCourt dataset, Judge Decoding accepts more valid tokens, achieving up to 9× faster generation speeds with maintained quality on the Llama-3.1 model family, demonstrating a promising approach for efficient large language model inference.

**Strengths:**

The paper is well-written. The argument is clear and reasonable. The proposed method shows effectiveness.

**Weaknesses:**

I do not see significant weakness.

But the sentence "To speed up inference in such a setting, Chen et al. (2023); Leviathan et al. (2023) concurrently propose speculative decoding (SD)" in the introduction section is incorrect.
At least, there is a much earlier one in 2018 that proposed speculative decoding: "Blockwise Parallel Decoding for Deep Autoregressive Models" by Mitchell Stern, Noam Shazeer, and Jakob Uszkoreit. The author skipped a large amount of literature on speculative decoding before 2023.

Authors are required to address this issue.

**Questions:**

NA

---

> ### Author Response · Authors · 2024-11-21
> **Author response**
>
> We thank the reviewer for the very positive feedback. We will address the very valid point of missing related works below:
>
>
> **Related works:** We agree with the reviewer and apologize for this oversight on our part. We added the following works on speculative decoding to the related works section:
> 1. *Blockwise Parallel Decoding for Deep Autoregressive Models*, Stern et al., 2021
> 2. *Instantaneous grammatical error correction with shallow aggressive decoding*, Sun et al., 2021
> 3. *Speculative Decoding: Exploiting Speculative Execution for Accelerating Seq2seq Generation*, Xia et al., 2022
> 4. *Accelerating Transformer Inference for Translation via Parallel Decoding*, Santilli et al., 2023
>
>
> We’re happy to further expand this section in case we are missing more relevant works.

---

### Official Review · Reviewer_yijH · 2024-11-04

**Soundness:** 4
**Presentation:** 4
**Contribution:** 4
**Rating:** 8
**Confidence:** 3

**Summary:**

The submission proposes Judge Decoding (JD), a variation of Speculative Decoding (SD) for LLMs. Like SD, JD uses a small draft model to propose output, and uses a larger model to check the output. The difference between SD and JD is in how the check is performed: where SD focuses on checking that the draft model's output is exactly like the larger model's (i.e., effectively per-token equality), JD uses a weaker criterion, based on the embeddings of the final layer of the larger (judge) model. The downside of this weaker criterion is that the output of the model is not guaranteed to be equivalent to the larger model's output anymore, but experiments indicate a significant potential speedup.

**Strengths:**

* Clear presentation, covering both technical details and high-level ideas
* Comprehensive experiments that emphasise the strengths of the proposed method
* Reported speedups seem significant, and seem like they could be very useful for practical serving of LLMs

**Weaknesses:**

* Some of the results (e.g., lack of agreement between different models / human drafts and models) seem to be obvious and do not help to make the argument
* The use of a disjunctive criterion (standard SD accepts or JD accepts) is a bit surprising, and left me wondering in what cases (and how often) JD rejects when SD accepts. The paper would be strengthened by investigating this more closely.
* Sect. 5.3 (OOD performance) is very important for deployment of the proposed method, and so it would be useful to expand on this. In particular, Sect. 4.1/"Dataset Curation" does not go into much detail into what distribution the training data follows, and which tasks are covered well. It would be helpful to understand whether there is a strong alignment between the TokenCourt training data and the considered evals.

**Questions:**

* Fig 2/Fig 3: It would be useful to report the number of proposed tokens as well, either explicitly or by presenting "ratio of accepted tokens" rather than an absolute number.
* One surprising result is not investigated: using Llama 405B to draft outputs of the 8B model on GSM8K (lower right of Fig. 2) seems to work less well than the other way around, even though the models are obviously well-aligned. I would have expected a basically equal result here (following the intuition that the larger model is effectively a superset of the smaller model), and was very surprised to see a lower score.

---

> ### Author Response · Authors · 2024-11-21
> **Author response**
>
> We thank the reviewer for the very positive feedback and the very helpful review. We will address the questions and weaknesses in the following:
>
> 1. **Some of the results are obvious:** We agree that it is not surprising that standard speculative decoding fails when very different models or human text are used as drafts, precisely due to the reliance on alignment. We think the more surprising or interesting part is that our method, judge decoding, does manage to accept a high number of tokens in these scenarios, as it relies now on the contextual quality. These experiments thus highlight the switch in the evaluation criterion: SD relies heavily on alignment, while JD evaluates quality. We hope this explains the inclusion of these more “obvious” experiments, as they are needed to enable later comparisons to our approach.
> 2. **Disjunctive criterion:** This is a great question and the choice seems to be a bit unnatural at first. We rely on the disjunctive criterion for the following reason:
> *If we replace a JD-rejected token with the most likely token according to the target model, we naturally end up with the disjunctive criterion.*
> We have also added a more detailed explanation in Appendix B.2 to clarify this further. If JD rejects a token, we replace this rejected token with the corresponding token produced by the target (just like in standard SD). Sometimes this target token is identical to the JD-rejected token. This can sometimes happen because JD is calibrated to rather err to reject tokens than to accept wrong ones. We thus “correct” the JD-rejected token with the same target token, and continue generating. Standard SD on the other hand would accept in this scenario (the proposed token is identical to the target token) and the course of sampling wouldn’t change; we would simply save computations by accepting the token already in this step, instead of adding it and re-generating draft tokens.
> Thus, as soon as we decide to replace rejected tokens with corresponding target tokens, we naturally end up with the disjunctive criterion. In order to make progress once a rejection is reached, we don’t see other alternatives beyond simply taking the corresponding target token. We hope this makes sense!
> In our experiments, however, we don’t observe this very often, JD accepted tokens are almost always a superset of the SD accepted ones. We still implement inference this way however to be most optimal.
> 3. **OOD performance:** The dataset largely consists of general input prompts from the Alpaca dataset [1] that were hand-filtered by us, which were not targeting specific evaluations but served to teach the judge the concepts of “correct” and “wrong” from many angles. A smaller percentage of prompts were created to more closely mimic evals, e.g. we added math questions to match performance on GSM8K, and we added coding examples to improve on HumanEval. The number of such additional prompts needed turned out to be very small, usually only 10-20 examples were needed to restore accuracy. There is also a high transfer between tasks; for ARC we included 20 prompts from its training set to teach the judge to evaluate multiple-choice answers. For MMLU (another multiple-choice dataset), there was no need to include further prompts as soon as ARC training prompts were added, as the notion of correct or wrong for MC questions of one task seems to be enough to transfer to others.
> In summary, there definitely is some alignment between the tasks we consider, and the dataset constructed, but we find that only a few prompts are needed for some tasks, while other tasks already end up being covered (e.g. MMLU and MT-Bench). In general, we expect the amount of new data needed to reduce significantly in the future as the dataset starts to grow and cover more and more aspects, i.e. the amount of completely new tasks (such as coding or math) will reduce quickly.
>
>
> 4. **Fig 2/Fig 3:** Since we cannot “speculate” with closed-source models or human text, we decided to produce full answers to all prompts and simply evaluate how many tokens are accepted until we reach the first rejection within the entire response. There was no natural choice for “number of draft tokens” and we thus decided to use full answers instead.
>
>
> 5. **Draft/Target switched:** This is an interesting observation! The rates are similar (6.3 vs 6.6) but slightly in favor of using Llama-405b as the draft and Llama-8b as the target, as the reviewer correctly points out. We think that Llama-405b might be better at mimicking the style of the outputs from Llama-8b the more it gets conditioned on its outputs (since it's a more capable model), compared to Llama-8b matching the style of Llama-405b. It thus seems reasonable to expect a better rate for Llama-405b as the draft model. We would be happy to discuss this further with the reviewer. We will update the paper to add this explanation.
>
>
> [1] *Stanford Alpaca: An Instruction-following LLaMA model*, Taori et al., 2023

---

### Public Comment · ~Weilin_Zhao1 · 2024-11-21
**Doubts About Baseline Speed in Table 1 Under the GPT-Fast Framework**

Ideally, framework changes should affect all methods **consistently**. This is because changing the framework is indeed altering the runtime difference between large and small models. **However**, the impact of switching frameworks on the speedup ratio is **not consistent** across the table.

Comparing Huggingface with GPT-Fast in Table 1

1) For the 405B model, STANDARD (5.3x `>` 1.7x), JUDGE (9.7x `>` 3.9x) and Medusa (6x `>` 1.9x) show similar trend, which shows consistency.
2) For the 70B model , STANDARD (1.5x `<` 1.7x), JUDGE (2.0x `<` 3.0x) follows the same trend but EAGLE (3.3x `>` 1.9x) does not, which is inconsistent.

The authors explain in the paper that EAGLE's slowdown in GPT-Fast is similar to the findings presented in EAGLE's original paper. However, we consider it is due to the suboptimal implementation of EAGLE in GPT-Fast.

To clarify, we suggest adding these baselines for the 70B model:

1. MEDUSA: In Table 1, Medusa is only reported in the 405B model but not reported in the 70B model.

2. STANDARD speculative decoding + lenience factor: [1] has already proposed using a lenience factor to relax the acceptance criteria for speculative sampling, which aligns with the motivation of this paper and should be included as a baseline.

[1] Leviathan, Yaniv, Matan Kalman, and Yossi Matias. "Fast inference from transformers via speculative decoding." International Conference on Machine Learning. PMLR, 2023.

---

> ### Author Response · Authors · 2024-11-24
> **Clarifications**
>
> Hi Weilin,
>
> Thank you for your interest and careful reading of our work! We want to clarify a few things:
>
> 1. **Medusa speedup:** The 6x for Medusa in HuggingFace is an upper bound, since 6 tokens could be maximally accepted if things would work perfectly with zero overhead. We don’t report a concrete number because [1] did not opensource their model and only ran it in (yet) another optimised framework and on different hardware, resulting in the 1.9x number we report. We mark it with an asterisk for these reasons. We thus don’t recommend comparing things directly using the 6x number.
> 2. **Reversing speedups:** We agree that at first it seems counter-intuitive that the trend reverses for the 70B target case, where Eagle speedups become worse in *gpt-fast*, while judge-decoding becomes faster. The driving factor in these speedup numbers is the discrepancy between draft and target speed, which changes in surprising ways when switching between frameworks.
> **In short:** 405B/8B latency gap reduces when compiling, while the gap of 70B/8B increases when compiling, thus leading to the reversed speedups.
> We will explain in more detail below:
>
>      - **8B/405B**: 405B is extremely slow in HF, leading thus to a large discrepancy to 8B even without compiling (in contrast to the 8B/70B setup). As a consequence our approach enjoys large speedups since we have a large discrepancy between target and draft. When compiling, both 8B and 405B become significantly faster in isolation (as expected). However, 405B enjoys even more speedup than 8B (we’re not entirely sure why but HF is probably not optimised for this size) , which thus leads to a **reduction** in terms of the speedup ratio, since the discrepancy between 8B and 405B is smaller now. Medusa or Eagle heads on the other hand don’t become much faster due to compilation since they’re very small models to begin with and not much overhead needs to be removed. In their cases, only the target model becomes significantly faster, reducing thus speedup numbers. This is consistent with results reported in Eagle [2], where speedup numbers reduced from 2.8x to 1.5x when compiling and quantising. [1] also finds a speedup of 1.9x in case of Medusa when compiling, which most likely is smaller than the one in HF.
>
>     - **8B/70B**: In HF, counter-intuitively, the latency gap between 70B and 8B is small. This is because both models are bottlenecked by CPU instructions at very small batch sizes. This is why SD and JD enjoy little speedups here; the draft is simply too slow compared to the target, as observed in many previous works for standard SD as well. This issue does not apply for Eagle; the very small module on top does run faster than the target even in HF and it thus dominates in this regime. When compiling the models, CPU-bound issues are removed and 8B runs significantly faster than 70B, leading to a **larger** latency gap. This leads to an improvement in the speedup numbers both for our JD approach and standard SD as well. This is consistent with results reported in the repo *gpt-fast* (https://github.com/pytorch-labs/gpt-fast), where 70B can be sped up by 2x using 8B as target, in contrast to the usual results that report 1.4x when using HF in this scenario (see e.g. [5]). Recall that in case of 405B, the target improved significantly more due to compilation than 8B did, which is why speedup numbers went down in this case. For Eagle on the other hand, again, compilation mainly benefits the target since the draft was already fast, leading thus to a reduction in speedup. Similarly, [1] find a speedup of 1.45x for Medusa for 70B, which again most likely is significantly smaller than speedups in HF (e.g. [3] reports a speedup of 2.83x in case of Vicuna-13B, there is unfortunately no larger model reported).
> 3. **Medusa for 70b**: This is a good point but we’re not aware of a publicly available version for Medusa in case of the 70B version. The same work by NVIDIA [1] that we cite for the 405B version does show speedup numbers also for 70B, which roughly are 1.45x when compiled. As expected based on prior work, the speedup is worse compared to our results for Eagle. We’re happy to further include this number too if this is helpful.
> Thank you for pointing out the useful baseline from [4]. We hope to include it in the next version of our paper.
>
> We’re happy to rectify any mistake if there are better known speedup numbers when compiling!
>
> [1] https://developer.nvidia.com/blog/low-latency-inference-chapter-1-up-to-1-9x-higher-llama-3-1-performance-with-medusa-on-nvidia-hgx-h200-with-nvlink-switch/
>
> [2] *EAGLE: Speculative Sampling Requires Rethinking Feature Uncertainty*, Li et al., 2024
>
> [3] *MEDUSA: Simple LLM Inference Acceleration Framework with Multiple
> Decoding Heads*, Can et al., 2024
>
> [4] *Fast inference from transformers via speculative decoding*, Leviathan et al., 2023
>
> [5] EAGLE-2: Faster Inference of Language Models with Dynamic Draft Trees, Li et al., 2024

---

> > ### Public Comment · ~Weilin_Zhao1 · 2024-11-27
> >
> > Thank you for your patient explanation. The ideas in the article are quite interesting, and I look forward to seeing the baseline further completed.

---

### Public Comment · ~Drew_Jin1 · 2025-04-15
**Inquiry About Open-Sourcing the Code**

Dear Authors,

I hope this message finds you well. I recently had the opportunity to go through your work on speculative decoding, and I must say it is truly impressive.

I was wondering if there are any plans to **open-source the code** related to this project. It would be incredibly beneficial for the community to explore, learn from, and potentially build upon your work. If the code is already available, could you kindly point me in the right direction?

Thank you for your time, and I look forward to hearing from you.

Best regards

---

### Meta-Review · Area_Chair_gigm · 2024-12-17

**Metareview:**

Summary: Different from the checking mode in the standard speculative sampling framework, this paper studies the problem of accelerating LLM inference by LLM-as-a-judge: asking the LLM itself whether the drafted content is valid or not. Experiments on Llama verify the effectiveness of the proposed method.

Strength:
1. The paper is well-written.
2. The experiments are sufficient.
3. The proposed method is effective by the experiments.

Weakness:
1. For tasks beyond QA, there might be no standard answer for the correctness. It is unclear how to use LLM-as-a-judge for such tasks, as there might be multiple "correct" answers.
2. The proposed framework changes the output text distribution of the target model. So it is not directly comparable with other distribution-preserving method, such as EAGLE and Medusa. The claimed speedup ratio might not be the only criteria to evaluate the effectiveness. One needs to take into account the trade-off between quality loss and speedup ratio.
3. LLM may have hallucination. LLM-as-a-judge might not always be reliable.

Reviewers are all positive about the paper. Therefore, AC are happy to recommend the paper for acceptance with spotlight. However, AC would not recommend the paper with an oral presentation given the above limitations, mostly raised by Reviewer Sf3Q.

**Additional Comments On Reviewer Discussion:**

Reviewer Sf3Q changes his/her score from 5 to 6 after the rebuttal, while the other three reviewers keep positive from the beginning. AC also notices a comment from Reviewer Sf3Q after the rebuttal: "It is clear that the authors understand and are forthcoming about the limitations of their novel approach but remain enthusiastic about the potential benefits." AC explains this as the fact that the authors understand and admit the limitations of this paper during the rebuttal phase. Given the positive scores by the other three reviewers and the limitations raised by Reviewer Sf3Q, AC would recommend the paper for acceptance with spotlight.

---

### Decision · Program_Chairs · 2025-01-22

Accept (Oral)